# Gradient Inversion of Multimodal Models

**Omri Ben Hemo** [* 1]  **Alon Zolfi** [* 1]  **Oryan Yehezkel** [1]  **Omer Hofman** [2]  **Roman Vainshtein** [2]  **Hisashi Kojima** [3]
**Yuval Elovici** [1]  **Asaf Shabtai** [1]

## Abstract

Federated learning (FL) enables privacy-preserving distributed machine learning by sharing gradients instead of raw data. However, FL remains vulnerable to gradient inversion attacks, in which shared gradients can reveal sensitive training data. Prior research has mainly concentrated on unimodal tasks, particularly image classification, examining the reconstruction of single-modality data, and analyzing privacy vulnerabilities in these relatively simple scenarios. As multimodal models are increasingly used to address complex vision-language tasks, it becomes essential to assess the privacy risks inherent in these architectures. In this paper, we explore gradient inversion attacks targeting multimodal vision-language Document Visual Question Answering (DQA) models and propose GI-DQA, a novel method that reconstructs private document content from gradients. Through extensive evaluation on state-of-the-art DQA models, our approach exposes critical privacy vulnerabilities and highlights the urgent need for robust defenses to secure multimodal FL systems. Project page at: https://AlonZolfi.github.io/GI-DQA/.

## 1. Introduction

Federated learning (McMahan et al., 2017) has emerged as a popular paradigm for privacy-preserving distributed machine learning. In FL, multiple clients collaboratively train a global model under the coordination of a central server over several rounds. During each round, clients update the local model using their private training data and transmit only the computed gradients to the server. The server aggregates

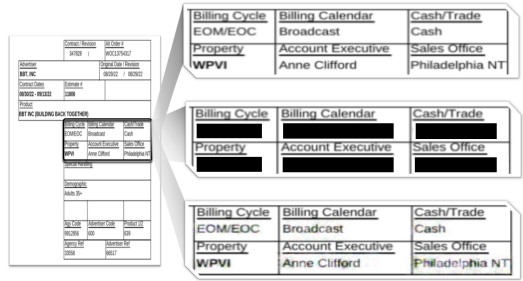

Figure 1: Gradient inversion attack on DQA models in FL. Left: document with private data from a benign client. Right: zoomed-in section shown in three versions (top to bottom): original, adversary's reference (public document template), and GI-DQA reconstruction—demonstrating private data recovery using only gradients and a document template.

these gradients to update the global model, reducing privacy risks by avoiding the exchange of raw data (Banabilah et al., 2022). This paradigm has been successfully applied in real-world scenarios, including Google Keyboard (Yang et al., 2018), Apple's Siri (Paulik et al., 2021), and critical domains such as healthcare (Thrasher et al., 2023) and finance (Shi et al., 2023).

Despite its privacy-preserving design, FL remains vulnerable to gradient inversion attacks (Du et al., 2024), which exploit the gradients shared by clients to reconstruct sensitive information from clients' private training data. Prior research has primarily focused on recovering private image data using optimization-based methods, including iterative optimization to minimize gradient differences (Zhu et al., 2019; Zhao et al., 2020), enhancements through cosine similarity and regularization (Geiping et al., 2020), and leveraging batch normalization for improved recovery in batches (Yin et al., 2021). More recent efforts have extended these attacks to complex architectures such as vision transformers (Lu et al., 2021; Hatamizadeh et al., 2022). While gradient inversion attacks have been extensively studied in unimodal tasks like image classification, their potential impact on multimodal applications remains unexplored.

Recently, multimodal models have gained significant attention for their ability to process and integrate diverse sources of information, including text, images, and spatial features,

---

[*]Equal contribution [1]Ben Gurion University of the Negev, Israel [2]Fujitsu Research of Europe [3]Fujitsu Limited. Correspondence to: Omri Ben Hemo / Alon Zolfi <{omribenh,zolfi}@post.bgu.ac.il>.

*Proceedings of the 42nd International Conference on Machine Learning*, Vancouver, Canada. PMLR 267, 2025. Copyright 2025 by the author(s).

enabling improved performance across a wide range of vision and language tasks (Wang et al., 2023). Among these, DQA models stand out for their focus on addressing document-related queries by leveraging multimodal capabilities. Broadly, DQA models can be categorized as either Optical Character Recognition (OCR)-based or OCR-free approaches. OCR-based models, such as the LayoutLM family (Xu et al., 2020a;b; Huang et al., 2022), rely on the extraction of textual information prior to processing and the combination of it with spatial and visual representations to achieve state-of-the-art performance. In contrast, OCR-free models, such as Donut (Kim et al., 2022), bypass external OCR modules by directly learning from document images. Compared to unimodal models, which process only a single type of data, multimodal models excel at capturing intricate cross-modality relationships, offering richer semantic and contextual understanding. This capability not only enhances their effectiveness in handling sophisticated applications, but also underscores their growing prominence in cutting-edge research. Despite their effectiveness, training multimodal models requires access to large, privacy-sensitive datasets that often contain confidential text and images. Centralizing such data for training poses privacy risks, regulatory concerns (*e.g.*, GDPR, HIPAA), and logistical challenges. FL offers a promising solution by enabling multimodal models to be trained collaboratively across decentralized data sources without exposing raw data (Nguyen & Karatzas, 2024).

Despite the increasing popularity of multimodal models, gradient inversion attacks on them remain significantly underexplored, particularly in decentralized learning environments such as FL. To address this gap, we present a comprehensive study of gradient inversion attacks targeting DQA models. DQA is particularly interesting due to the sensitive and private nature of the documents it processes, such as medical records, financial statements, and legal contracts, making privacy concerns especially critical in real-world applications. Specifically, we propose GI-DQA, a novel method that reconstructs sensitive document content (*e.g.*, personal identifiable information) from shared gradients in an FL setup. To reflect realistic attacker capabilities, we assume access to standardized document templates, which are common in real-world domains (*e.g.*, invoices, and medical forms). These templates follow fixed layouts with only a few variable fields, allowing the attacker to focus reconstruction efforts on regions likely to contain private information, as shown in Figure 1. By leveraging the unique structure of multimodal models, GI-DQA exposes critical privacy vulnerabilities inherent in these models, providing deeper insight into the risks associated with multimodal architectures and emphasizing the need for robust countermeasures to protect sensitive information. DQA-specific gradient inversion presents unique challenges: *(i)* reconstructing small-sized

words in document images is significantly harder than recovering large objects in natural images. Unlike large-scale structures that provide strong spatial cues, fine-grained text is densely packed, making even slight variations in gradients highly disruptive to reconstruction, *(ii)* text-based gradients are inherently sparse, which limits the amount of recoverable information. This sparsity makes it difficult to distinguish meaningful features from noise, further complicating precise recovery. These factors make DQA-specific gradient inversion uniquely challenging and require more advanced techniques than those used in standard vision-based attacks.

To evaluate the effectiveness of our proposed method, we conduct extensive experiments on state-of-the-art DQA models, including both OCR-based and OCR-free architectures. Our evaluation demonstrates the feasibility of reconstructing private data with high fidelity, shedding light on the weaknesses of existing privacy-preserving mechanisms in multimodal FL setups. For example, on the Donut model, our attack is able to perfectly reconstruct 70.1% of the documents' words.

Our contributions can be summarized as follows:

- To the best of our knowledge, we present the first gradient inversion attack on multimodal models, introducing a novel method specifically tailored for multimodal DQA models.
- A thorough evaluation of privacy vulnerabilities in state-of-the-art OCR-based and OCR-free DQA architectures.
- Insights into the unique challenges of performing gradient inversion on multimodal tasks, highlighting critical areas for future research.
- An exploration of potential defense mechanisms to mitigate privacy risks, offering a roadmap to enhance the robustness of multimodal models in FL systems.

## 2. Background & Related Work

### 2.1. Gradient Inversion in Federated Learning

Gradient inversion attacks in FL aim to reconstruct private data by exploiting shared gradients. Prior attacks can be categorized by: *(i)* server's trustworthiness: adversaries located on the server may be *honest-but-curious* (Zhu et al., 2019; Zhao et al., 2020; Geiping et al., 2020; Hatamizadeh et al., 2022; Lu et al., 2021), passively analyzing gradients without interfering, or *malicious* (Fowl et al., 2021; Chu et al., 2022; Pasquini et al., 2022; Fowl et al., 2022; Wen et al., 2022), actively disrupting the training process; and *(ii)* attack strategy: in *optimization-based* (Geiping et al., 2020; Yin et al., 2021; Hatamizadeh et al., 2022; Gupta et al., 2022; Lu et al., 2021) attacks, the ground-truth samples are approximated through iterative refinement, while in *analytic-based* (Fan et al., 2020; Zhu & Blaschko, 2020)

attacks, systems of equations are solved between gradients and inputs to precisely reconstruct the data. In this work, we address the more challenging and realistic honest-but-curious setting, leveraging a combination of optimization- and analytic-based strategies.

Most prior research has focused on recovering private image data through optimization-based approaches. Zhu et al. (2019) (DLG) first formulated the attack as an iterative optimization problem, where attackers restore data samples by minimizing the distance between shared gradients and dummy gradients. Zhao et al. (2020) (iDLG) proposed extracting the label of a single sample from the gradients, which further improved the attack. Geiping et al. (2020) (Inverting Gradients) reconstructed higher-resolution images from ResNet-based models (He et al., 2016) by modifying the distance metric to cosine similarity and incorporating total variation as a regularization term. Yin et al. (2021) primarily focused on recovery for larger batch sizes. Their method assumes access to normalization (BN) statistics, which enabled partial image recovery at larger batch sizes. Hatamizadeh et al. (2022) (GradViT) and Lu et al. (2021) (APRIL) extended these attacks to vision transformers.

These works primarily address unimodal classification models, where reconstructed images typically contain a single prominent object. In contrast, our study focuses on an under-explored domain: the reconstruction of visual data from multimodal DQA models, where images consist of numerous fine-grained textual elements, introducing unique challenges for inversion. A recent related work (Liu et al., 2024) considers gradient inversion in a multimodal FL setting; however, it employs independently trained, modality-specific models without shared representations, coordinated through mutual knowledge distillation. This approach fundamentally differs from our setting, which involves deeply fused, jointly trained architectures typical of modern multimodal models.

## 2.2. Document Visual Question Answering

DQA has emerged as a prominent area of interest in the machine learning research community due to its ability to answer natural language questions by extracting meaningful information from documents. The importance of efficient management of document workflows spans multiple industries, including banking, insurance, public administration, impacting virtually all aspects of business operations. A key differentiating factor among DQA models is their reliance on OCR technology. OCR-based models, such as the LayoutLM family (Xu et al., 2020a;b; Huang et al., 2022), TILT (Powalski et al., 2021), and UDOP (Tang et al., 2023), use OCR to integrate textual, layout, and visual features, achieving impressive performance. In contrast, OCR-free models adopt end-to-end architectures that bypass external OCR modules. These models, including Donut (Kim et al., 2022), Pix2Struct (Lee et al., 2023), and Dessurt (Davis

et al., 2022), incorporate reading-oriented pretraining objectives to directly process document images, offering competitive performance while reducing dependency on OCR tools. Both OCR-based and OCR-free approaches highlight the diverse methodologies in DQA, addressing trade-offs between interpretability, robustness, and performance across various document understanding tasks.

## 3. Method

### 3.1. Preliminaries

**DQA Federated Learning.** A FL framework for DQA consists of a server that coordinates $K$ clients, each with a private dataset $D_k$ of $N$ samples $(x, y)$, where $x$ represents the input data (visual document $x_D$ and question $x_Q$), and $y$ the target (corresponding answer). Clients collaboratively train a global DQA model $f_\theta$ with weights $\theta$ without exchanging raw data. In each round $t$, the server sends the current global model $f_{\theta_t}$ to the clients for local training, for a total of $T$ rounds. Each participating client computes the average gradient $\nabla_k \theta_t$ on its local data using the current global model:

$$\nabla_k \theta_t = \mathbb{E}_{(x,y) \in D_k}[\nabla \mathcal{L}(\theta_t; x, y)] \quad (1)$$

where $\mathcal{L}(\cdot)$ represents the standard DQA training loss. Finally, the server aggregates the collected gradients and updates the global model as follows:

$$f_{\theta_{t+1}} \leftarrow f_{\theta_t} + \eta \sum_{k=1}^{K} \nabla_k \theta_t \quad (2)$$

where $\eta$ is the learning rate.

**Gradient Inversion.** Given a gradient $\nabla_k \theta_t$ computed by client $k$ using the original data $(x, y) \in D_k$, an adversary generates randomly initialized data $(\hat{x}, \hat{y})$ which are iteratively updated to approximate $(x, y)$ by solving the following optimization problem:

$$\hat{x}^* = \arg\min_{\hat{x}} d(\nabla \mathcal{L}(\theta, \hat{x}, \hat{y}), \nabla_k \theta_t) + \sum_j \alpha_j \mathcal{R}_j \quad (3)$$

where $d$ is a distance metric (*e.g.*, euclidean distance), $\mathcal{R}$ represents a regularization term (*e.g.*, total variation (Chambolle et al., 2010)), and $\alpha$ is a weighting factor. Note that an adversary can perform the attack in any iteration $t \in T$.

### 3.2. Threat Model

**Adversary's Capabilities.** We consider an honest-but-curious adversary (Section 2.1) who has access to the gradients returned by an arbitrary client $k$, either by being physically located on the server or by eavesdropping on the communication between the central server and the client. At iteration $t$, the adversary has access to: *(i)* the gradients

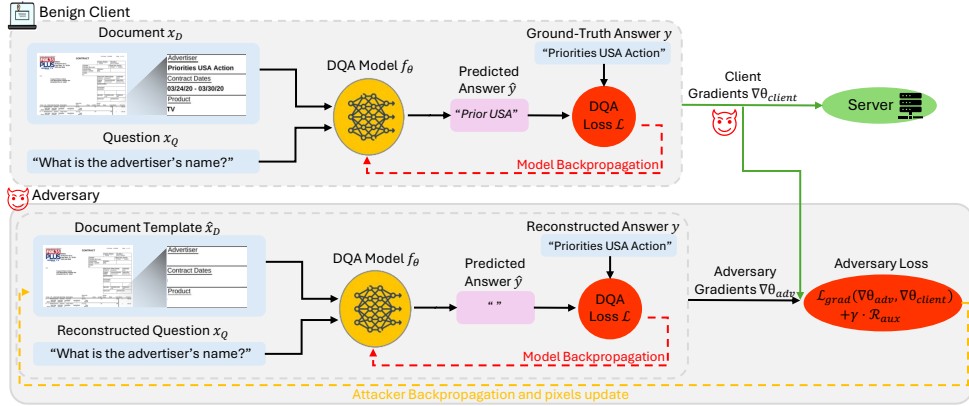

Figure 2: Overview of the FL setup with a benign client and an honest-but-curious attacker. The benign client computes gradients on their private document and shares them with the server, where the attacker is positioned. The attacker, equipped with a document template (without private data), computes their own gradients and optimizes a loss function comprising gradient matching and auxiliary priors to reconstruct the client's document.

$\nabla_k \theta_t$ shared by the client, *(ii)* the current model parameters $\theta_t$, and *(iii)* a collection of publicly available document image templates (*e.g.*, healthcare forms), which represent standard formats with private fields (such as names, dates, or amounts) left empty. These templates provide additional structural and contextual information that closely resemble the input documents, significantly narrowing the adversary's search space. This allows the adversary to focus on specific regions of interest within the documents, such as fields where private information is typically located. In many document-based applications, such as invoices, medical forms, boarding passes, or receipts, the layout and structure are highly standardized across users, with only a small number of sensitive fields varying. Therefore, this setup is practically grounded and reflects realistic scenarios. Figure 2 presents an overview of the attack pipeline.

**Adversary's Objective.** The adversary aims to reconstruct the private fields contained within the client's documents. These private fields may include personally identifiable information (PII), such as names (*e.g.*, patients), ID numbers (*e.g.*, SSNs), financial details (*e.g.*, account balances), and sensitive textual data (*e.g.*, medical diagnoses). The adversary seeks to exploit both the structure of the document (*i.e.*, templates) and contextual information encoded in the shared gradients. This combination of prior knowledge and gradient leakage significantly increases the likelihood of a privacy breach, posing a serious threat to client confidentiality.

### 3.3. Multimodal Gradient Inversion Attack

In this section, we describe our proposed approach, which consists of two components: document reconstruction and text reconstruction. The document reconstruction follows an optimization-based approach, while the text reconstruction employs an analytic-based approach.

Prior work on image classification models has demonstrated that using the ground truth label $y$ (rather than a randomly initialized one) significantly improves the quality of reconstruction (Zhao et al., 2020). This improvement is achieved by leveraging the negative sign traces of the gradient in the classification head. The enhancement arises because the computed gradients depend on the loss function, which is directly influenced by the ground-truth label, thereby reducing the distance between the adversary's and client's gradients and simplifying the optimization process. However, in our case, as we employ question-answering models, this approach is not applicable. Furthermore, the input $x$ in our setting comprises both the document $x_D$ and its corresponding question $x_Q$. The input question $x_Q$ also affects the calculated gradients, as it provides contextual information that guides the model to understand the relationship between the document $x_D$ and the target answer $y$. Consequently, reconstructing both the question $x_Q$ and the answer $y$ is critical for the success of the attack (discussed in Section 4.2.2). To achieve this, we build on previous works that propose analytic-based methods to first extract the question-answer pairs (Petrov et al., 2024; Deng et al., 2021; Balunovic et al., 2022). Once these pairs are reconstructed, we then optimize the document reconstruction to refine the overall results.

#### 3.3.1. DOCUMENT RECONSTRUCTION

For visual text reconstruction, we propose the following optimization objective:

$$\hat{x}^* = \arg\min_{\hat{x}} \mathcal{L}_{\text{grad}}(\nabla \mathcal{L}(\theta, \hat{x}, \hat{y}), \nabla_k \theta_t) + \varphi(t)\mathcal{R}_{\text{aux}}$$

$$\mathcal{R}_{\text{aux}} = \alpha_{\text{txt}}\mathcal{R}_{\text{txt}} + \alpha_{\text{gau}}\mathcal{R}_{\text{gau}} + \alpha_{\text{TV}}\mathcal{R}_{\text{TV}}$$

$$(4)$$

where $\mathcal{L}_{\text{grad}}$ represents the matching loss, and $\mathcal{R}_{\text{aux}}$ is a set of auxiliary priors. The scheduling term $\varphi(t)$ dynamically

adjusts the influence of the auxiliary priors throughout the optimization process. We incorporate three auxiliary priors to improve reconstruction fidelity: *(i)* a visual text prior based on the Laplacian filter, which enhances fine-grained text boundaries, *(ii)* a Gaussian prior, which encourages spatial smoothness and reduces high-frequency noise, and *(iii)* a total variation (TV) prior, which combines spatial smoothness with a channel-wise consistency constraint that encourages grayscale reconstructions, aligning with the typical appearance of text and background in documents. In the remainder of this section, we describe each prior and scheduler in detail.

**Gradient Matching.** The gradient matching loss serves as the primary component of the optimization, minimizing the discrepancy between the client's gradients and those of the adversary. Previous works have used distance metrics such as Euclidean distance and cosine similarity. In our approach, we combine both metrics for improved effectiveness: Euclidean distance is used to account for the magnitude of the gradient vectors, while cosine similarity attends to their directional alignment. This loss is defined as:

$$\mathcal{L}_{\text{grad}} = \sum_l \mathcal{L}_{\text{MSE}}\left(\nabla^l \mathcal{L}(\theta, \hat{x}, \hat{y}), \nabla_k^l \theta_t\right) + \\ \mathcal{L}_{\text{CS}}\left(\nabla^l \mathcal{L}(\theta, \hat{x}, \hat{y}), \nabla_k^l \theta_t\right) \quad (5)$$

where $l$ denotes the $l$-th layer, $\mathcal{L}_{\text{MSE}}$ is the mean squared loss, and $\mathcal{L}_{\text{CS}}$ is the cosine similarity loss.

**Visual Text Prior.** Accurately reconstructing small-sized words ($\sim$1% of the full image) is a highly challenging task due to their minimal contribution to the overall gradient and their low visibility in the reconstructed image. To address this, we introduce a new text prior based on the Laplacian filter (Paris et al., 2011). The Laplacian filter is a second-order derivative filter that emphasizes regions of rapid intensity changes in an image, such as edges and fine-grained details. It is particularly effective for extracting high-frequency features, such as the boundaries of text and intricate patterns, which are crucial for reconstructing specific word details. The discrete form of the Laplacian filter measures the sum of curvature in both the horizontal and vertical directions, which is expressed as:

$$\nabla^2 \hat{x_D}(i,j) = \frac{\partial^2 \hat{x_D}}{\partial i^2} + \frac{\partial^2 \hat{x_D}}{\partial j^2} \quad (6)$$

where $\frac{\partial^2 \hat{x_D}}{\partial i^2}$ and $\frac{\partial^2 \hat{x_D}}{\partial j^2}$ are the second derivatives of the image intensity $\hat{x_D}$ with respect to spatial coordinates $i$ and $j$. In practice, the Laplacian filter is implemented using a

convolution operation with a fixed kernel:

$$\mathcal{R}_{\text{txt}}(\hat{x_D}) = \frac{1}{|\hat{x_D}|} \sum_i \sum_j |(\mathbf{K}_{\text{lap}} * \hat{x_D})(i,j)|$$

$$\mathbf{K}_{\text{lap}} = \begin{bmatrix} 0 & -1 & 0 \\ -1 & 4 & -1 \\ 0 & -1 & 0 \end{bmatrix} \quad (7)$$

where $\mathbf{K}_{\text{Lap}}$ is the 4-neighbor Laplacian kernel, $(\mathbf{K}_{\text{lap}} * \hat{x_D})(i,j)$ is the convolution operation at a specific pixel $(i,j)$, and $|\hat{x_D}|$ is the total number of pixels in the image.

**Gaussian Prior.** To enhance the reconstruction quality of fine-grained details, we introduce a Gaussian prior, which leverages the properties of a Gaussian smoothing filter to regularize the optimization process. The Gaussian prior enforces spatial consistency by penalizing high-frequency noise while preserving key structures, such as edges and text regions, acting as a low-pass filter. Similar to the Laplacian filter, the Gaussian filter is implemented using a convolution operation with a Gaussian kernel:

$$\mathcal{R}_{\text{gau}}(\hat{x_D}) = \frac{1}{|\hat{x_D}|} \sum_i \sum_j |(\mathbf{K}_{\text{gau}} * \hat{x_D})(i,j) - \hat{x_D}(i,j)|$$

$$\mathbf{K}_{\text{gau}}(i,j) = \frac{1}{2\pi\sigma^2} \exp\left(-\frac{i^2 + j^2}{2\sigma^2}\right) \quad (8)$$

By applying the Gaussian filter to the reconstructed image and computing the difference with the unfiltered one, this prior quantifies deviations in smoothness. This encourages the optimization to focus on preserving the natural smoothness of the input image while avoiding overfitting to noise or artifacts. The Gaussian prior is particularly effective for ensuring smoother transitions in reconstructed document regions, facilitating a more realistic and accurate reconstruction. Compared to the commonly used total variation prior (Chambolle et al., 2010), which focuses on differences between neighboring pixel values, the Gaussian prior applies a smoothing filter globally across larger regions, reducing variations uniformly, including at edges.

**Total Variation Prior.** The TV prior is widely used in image reconstruction tasks to enforce smoothness while preserving important structures. Following prior works (Geiping et al., 2020; Hatamizadeh et al., 2022), we adopt the standard spatial TV formulation, which measures the variation between neighboring pixels in the spatial dimensions of an image (*i.e.*, width and height). For an image $\hat{x_D}$ with spatial dimensions $(i,j)$, the spatial TV is defined as:

$$\mathcal{R}_{\text{TV-S}}(\hat{x_D}) = \sum_{i,j} |\hat{x}_{D(i+1,j)} - \hat{x}_{D(i,j)}| + \\ |\hat{x}_{D(i,j+1)} - \hat{x}_{D(i,j)}| \quad (9)$$



Figure 3: Color palette where the RGB channels have identical intensity (R=G=B). The palette ranges from black (R=G=B=0) to white (R=G=B=255), illustrating colors that vary solely in brightness without hue or saturation. This reflects the effect of the TV channel prior, which enforces RGB consistency and promotes grayscale reconstructions, aligning with the typical appearance of text and backgrounds in documents.

In addition to spatial smoothness, we introduce a channel TV prior, which operates on the color channels (*i.e.*, red, green, and blue) at each pixel. The channel TV penalizes variations in color intensity within a pixel across the RGB channels, helping suppress chromatic artifacts and reducing color noise. For an image where the RGB channels are forced to have equal intensity (R=G=B), the resulting colors belong to the grayscale spectrum. This spectrum consists of shades that vary only in brightness, ranging from black (R=G=B=0) to white (R=G=B=1), without any hue or saturation, as shown in Figure 3. The channel TV prior leverages this property to encourage grayscale consistency across the reconstructed image, aligning with the nature of text reconstruction tasks, where content (*e.g.*, text or background) is predominantly black, white, or gray. The channel TV is defined as:

$$\mathcal{R}_{\text{TV-C}}(\hat{x_D}) = \sum_{i,j} |\hat{x}_{D(i,j,R)} - \hat{x}_{D(i,j,G)}| +$$
$$|\hat{x}_{D(i,j,G)} - \hat{x}_{D(i,j,B)}| + |\hat{x}_{D(i,j,B)} - \hat{x}_{D(i,j,R)}| \quad (10)$$

Combining both spatial and channel priors, the overall TV prior is expressed as:

$$\mathbf{R}_{\text{TV}}(\hat{x_D}) = \mathcal{R}_{\text{TV-S}}(\hat{x_D}) + \mathcal{R}_{\text{TV-C}}(\hat{x_D}) \quad (11)$$

By combining spatial and channel priors, the overall TV prior enforces both local smoothness across neighboring pixels and consistency in color channels. The inclusion of the channel TV prior ensures that reconstructed images adhere to the grayscale spectrum, effectively promoting natural and visually coherent reconstructions that align with the characteristics of text or background.

**Priors Scheduler.** Unlike prior work (Hatamizadeh et al., 2022) that delayed the activation of auxiliary priors until later stages of optimization to avoid suboptimal convergence, we propose a scheduler that assigns high weights to priors early in training. This approach stems from our observation that auxiliary priors play a critical role in stabilizing the early stages of optimization, where the synthesized inputs are far from convergence. By emphasizing priors early

on, our scheduler ensures that the optimization process is guided toward realistic reconstructions from the start. We adopt an exponential decay strategy for $\varphi(t)$, defined as $\varphi(t) = \alpha \exp(-\lambda t)$, where $\alpha$ is the initial weighting coefficient and $\lambda$ is the scaling factor that determines the decay rate. This allows the contribution of auxiliary priors to diminish gradually as optimization progresses, letting the gradient matching loss take precedence in later stages for fine-tuning. This shift enables the priors to shape the overall structure early on, while the gradient matching loss refines details as optimization converges.

## 4. Evaluation

### 4.1. Evaluation Setup

**Models.** We use state-of-the-art DQA models, covering both OCR-free and OCR-based models: *(i)* Document Understanding Transformer (**Donut**) (Kim et al., 2022) - an OCR-free model that employs a transformer-based encoder-decoder architecture. Consists of a vision encoder (Swin Transformer (Liu et al., 2021)) and a text decoder (BART (Lewis, 2019)). We use the base-size model with 156M parameters. *(ii)* **LayoutLMv3** (Huang et al., 2022) - an OCR-based model and the latest version of the LayoutLM family models (Xu et al., 2020a), which employs a transformer-based architecture with visual patch embeddings and incorporates textual input along with bounding box coordinates for spatial context (*i.e.*, OCR). We use the base-size model, which consists of 133M parameters.

**Datasets.** We use the PFL-DocVQA (Tito et al., 2024) dataset, designed to perform DocVQA in a FL environment, with the aim of exposing privacy leakage issues in a realistic scenario. It comprises invoice document images and a set of question/answer pairs. The documents contain sensitive data of the invoice provider identity, including provider name, address, and other sensitive fields. For our experiments, we use a subset of the original dataset containing 395 documents. The subset includes 90 templates, each with approximately five distinct documents (the sensitive data differs between the documents of the same template).

**Metrics.** In our evaluation, performance is measured with metrics commonly used in the visual gradient inversion domain (Hatamizadeh et al., 2022; Lu et al., 2021; Geiping et al., 2020): *(i)* **Peak Signal-to-Noise Ratio (PSNR)** - quality of reconstructed images by quantifying the ratio between the maximum possible signal value and the distortion introduced by the reconstruction. *(ii)* **Cosine similarity in the Fourier space (FFT$_{\text{2D}}$)** - similarity between the original and reconstructed images in the frequency domain. *(iii)* **Mean Squared Error (MSE)** - average squared difference between corresponding pixels in the original and reconstructed images.

Table 1: Main results. Evaluation of our attack and prior works on the Donut and LayoutLMv3 models. ↓ indicates that lower values are better, and ↑ indicates that higher values are better. Bold indicates superior results.

| Method | Donut | | | | | LayoutLMv3 | | | | |
|---|---|---|---|---|---|---|---|---|---|---|
| | PSNR ↑ | FFT$_{2D}$ ↓ | MSE ↓ | Binary ↑ | Fuzz Ratio ↑ | PSNR ↑ | FFT$_{2D}$ ↓ | MSE ↓ | Binary ↑ | Fuzz Ratio ↑ |
| Random | 6.989 | 0.113 | 96.591 | 0% | 0 | 7.041 | 0.124 | 105.169 | 0.9% | 0 |
| DLG (Zhu et al., 2019) | 6.984 | 0.110 | 96.574 | 0% | 0.001 | 7.015 | 0.127 | 105.185 | 1.1% | 0 |
| iDLG (Zhao et al., 2020) | 6.984 | 0.110 | 96.574 | 0% | 0.001 | 7.015 | 0.127 | 105.185 | 1.1% | 0 |
| Inverting Gradients (Geiping et al., 2020) | 11.386 | 0.042 | 95.314 | 0% | 0.006 | 10.596 | 0.044 | 98.281 | 0% | 0 |
| APRIL (Lu et al., 2021) | 13.652 | 0.027 | 85.329 | 5.8% | 0.283 | 9.825 | 0.060 | 98.946 | 0.6% | 0 |
| GI-DQA (Ours) | **24.194** | **0.003** | **60.403** | **70.1%** | **0.909** | **22.713** | **0.005** | **72.508** | **82.1%** | **0.681** |

Table 2: Effect of different loss components on the reconstruction quality. ↓ indicates that lower values are better, and ↑ indicates that higher values are better.

| Loss Component | Reconstruction Metrics | | | | |
|---|---|---|---|---|---|
| | PSNR ↑ | FFT$_{2D}$ ↓ | MSE ↓ | Binary ↑ | Fuzz Ratio ↑ |
| $\mathcal{L}_{grad}$ | 17.872 | 0.012 | 75.008 | 42.6% | 0.759 |
| $+\mathcal{R}_{TV\text{-}C}$ | 22.555 | 0.006 | 63.823 | 61.0% | 0.856 |
| $+\mathcal{R}_{TV\text{-}S}$ | 24.540 | 0.003 | 58.063 | 70.2% | 0.894 |
| $+\mathcal{R}_{txt}$ | 23.855 | 0.003 | 61.655 | 70.6% | 0.913 |
| $+\mathcal{R}_{gau}$ | 24.194 | 0.003 | 60.403 | 70.1% | 0.909 |

$\mathcal{L}_{grad}$    $+\mathcal{R}_{TV\text{-}C}$    $+\mathcal{R}_{TV\text{-}S}$    $+\mathcal{R}_{txt}$    $+\mathcal{R}_{gau}$    Original

We also include metrics specifically designed for the DQA domain and visual text reconstruction. To achieve this, we use Trocr (Li et al., 2023), an OCR model to extract the reconstructed text, and measure: *(i)* **Exact Match (Binary)** - percentage of text segments that are perfectly reconstructed, reflecting the attack's ability to recover textual information without any character-level errors. *(ii)* **Fuzz Ratio (FR)** - A similarity score based on character-level differences using fuzzy string matching, allowing for a more flexible evaluation of partial reconstructions where minor distortions or OCR errors may still preserve semantic meaning. This approach evaluates the attack's effectiveness in an automated end-to-end scenario.

**Implementation Details.** The optimized pixels in the reconstructed document image are randomly initialized and updated using the Adam optimizer with an initial learning rate of 2.0, applying exponential decay with a rate of $\lambda = 0.999$ over 5,000 iterations. The auxiliary loss terms (Equation 4) are weighted using the coefficients $\alpha_{txt} = 0.1$, $\alpha_{gau} = 0.01$, and $\alpha_{TV} = 0.1$. These values were selected using the grid search approach over the values $\{0, 0.001, 0.01, 0.1, 1\}$, optimizing for PSNR performance.

### 4.2. Results

Here, we present the results for our proposed attack. We report additional results in Appendix A.1 and provide examples of reconstructed documents in Appendix A.3.

### 4.2.1. ATTACK EFFECTIVENESS

Table 1 presents the results of our proposed attack in comparison to prior works, including DLG (Zhu et al., 2019), iDLG (Zhao et al., 2020), Inverting Gradients (Geiping et al., 2020), and APRIL (Lu et al., 2021). As shown, existing methods designed for unimodal models fail to reconstruct meaningful content when applied to multimodal architectures. DLG and iDLG attack performance is identical to that of a random baseline. Even APRIL, which is designed for transformer-based models, lacks the capability to capture the intricate interactions between visual and textual modalities, leading to incomplete or incoherent reconstructions. These methods struggle to extract fine-grained document features, as they are not designed to handle the fusion of structured text and image embeddings present in multimodal systems. In contrast, our proposed attack effectively exploits the unique properties of multimodal DQA models, leveraging both pre-fusion gradients (explained in Section 5) and cross-modal dependencies to reconstruct sensitive document content with significantly higher fidelity. Furthermore, our approach is specifically designed to better fit the domain of visual text, ensuring more accurate recovery of structured textual elements that are critical in documents. For example, on the Donut model, our approach accurately reconstructs 70.1% of the set of words in all the documents combined, compared to 5.8% in APRIL, demonstrating its effectiveness in recovering fine-grained textual details. Examples are shown in Figure 4.

### 4.2.2. ABLATION STUDIES

**Priors Effect.** We analyze the contribution of each training loss term by evaluating its impact on reconstruction quality. Table 2 presents these results, providing quantitative comparisons alongside qualitative examples to illustrate their effectiveness. We observe that while optimizing $\mathcal{L}_{grad}$ alone enables partial reconstruction, it struggles to eliminate the random noise introduced by the adversary's document initialization, achieving a binary match of 42.6%. Introducing the channel-wise total variation prior $\mathcal{R}_{TV\text{-}C}$ yields a substantial improvement, increasing the binary accuracy to 61.0% and the PSNR to 22.56 by promoting grayscale consistency and reducing chromatic noise. Adding spatial TV

Figure 4: Examples of zoomed-in sections of documents where private information is reconstructed using different gradient inversion techniques on the Donut model. Censored represents the document with redacted private data, which serves as the adversary's initial reference before optimization.

prior $\mathcal{R}_{TV\text{-}S}$ further enhances smoothness across neighboring pixels, leading to the best overall PSNR of 24.54, which indicates its strong contribution to local structural coherence. The text prior $\mathcal{R}_{txt}$ slightly improves the binary match and fuzz ratio, sharpening edges and enhancing text legibility. The Gaussian prior $\mathcal{R}_{gau}$ helps suppress the remaining high-frequency noise, leading to an improved PSNR over $\mathcal{R}_{txt}$, stabilizing the reconstruction, and maintaining performance gains across all metrics.

**Question-Answer Effect.** As discussed in Section 3.3, the effectiveness of document reconstruction depends on prior access to the question-answer (QA) pair. Table 3 evaluates the attack's performance under different attacker knowledge scenarios: *(i)* black-box setting – both the question and answer are unknown (random tokens are used). *(ii)* gray-box setting – only one of the question or answer is known. *(iii)* white-box setting – both the question and answer are accurately reconstructed and used in the attack. The results demonstrate that document reconstruction is infeasible without first recovering both the question and answer, even when one component is known. In contrast, when the question-answer pair is fully known, the attack significantly improves, enabling the successful reconstruction of private document details. This reliance on the question-answer pair highlights a unique characteristic of multimodal models, where the fusion of different modalities plays a crucial role in preserving and extracting information. Unlike unimodal attacks that rely solely on direct feature recovery, multimodal gradient inversion depends on cross-modal interactions, making the reconstruction process inherently dependent on both textual and visual components. Although the answer in the QA pair may contain a piece of personally identifiable in-

Table 3: Impact of question and answer knowledge on document reconstruction ability. ↓ indicates that lower values are better, and ↑ indicates that higher values are better.
*Original question and answer were replaced with random tokens.

| Known | | Reconstruction Metrics | | | | |
|---|---|---|---|---|---|---|
| Question | Answer | PSNR ↑ | FFT$_{2D}$ ↓ | MSE ↓ | Binary ↑ | Fuzz Ratio ↑ |
| ✘ | ✘ | 5.237 | 0.196 | 87.212 | 0.8% | 0 |
| ✘ | ✔ | 6.956 | 0.124 | 88.743 | 0.7% | 0 |
| ✔ | ✘ | 5.240 | 0.195 | 87.207 | 1.0% | 0 |
| ✔ | ✔ | 24.194 | 0.003 | 60.403 | 70.1% | 0.909 |
| ✔* | ✔* | 24.439 | 0.002 | 59.677 | 71.9% | 0.920 |

formation (PII), it typically reflects only a small portion of the overall sensitive content. Additional PII fields, unmentioned in the QA, can still be inferred from the gradients (further discussed in the following section). Interestingly, our findings reveal that the semantic content of the question and answer is not critical to the attack's success. Instead, effective document reconstruction relies on the adversary having access to the exact token identities and their positions used by the client, regardless of their meaning. To demonstrate this, we replaced the original QA pair with random tokens of the same length and structure. The resulting reconstructions showed nearly identical quality across all metrics, as shown in Table 3 (bottom row). This indicates that gradient inversion exploits the precise token representations encoded in the model's gradients, rather than their semantic interpretation.

**Training Stage Effect.** We conduct a systematic evaluation of the attack's effectiveness at different stages of model training to understand how the evolution of learned representations influences reconstruction quality. Our findings reveal a clear trend: as the model becomes more optimized, inver-

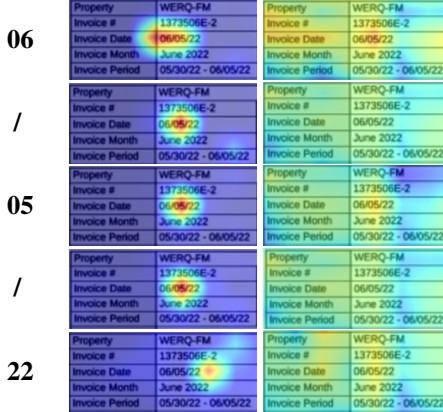

Figure 5: Cross-attention maps from the decoder for the input "06/05/22" split into tokens in a fine-tuned model (left) and a randomly initialized model (right). The fine-tuned model exhibits sharply focused attention, while the randomly initialized model attends to broader regions, enabling more recoverable information during inversion.

sion quality degrades significantly (full results can be found in Appendix A.1.2). This suggests that, early in training, the model encodes a broader and more spatially distributed set of features, as its cross-attention tends to be more diffuse and less focused on specific regions. In contrast, as training progresses, attention and token representations become increasingly concentrated on task-relevant areas, particularly those directly associated with the question and answer, thereby narrowing the gradient signal, as shown in Figure 5. This focused representation reduces the diversity of information available in the gradients, making it more difficult to reconstruct PII beyond the QA scope. This observation also helps explain why our attack remains effective even when QA tokens are replaced with random tokens, as long as the adversary has access to the same input tokens.

## 5. Discussion

A key aspect of understanding privacy risks in multimodal models is identifying which gradients are most susceptible to inversion. In our analysis, we find that only a subset of the gradients meaningfully contribute to successful reconstruction. Multimodal vision-language models typically follow three stages: (i) extracting features from visual and textual inputs, (ii) fusing these modalities, and (iii) generating the output. Our findings show that gradients from the pre-fusion stage, where the model still processes each modality independently, carry the most informative signal for reconstruction. In contrast, post-fusion gradients are significantly less revealing. In Donut, relying only on post-fusion gradients makes reconstruction slower and less accurate, while in LayoutLMv3, reconstruction fails entirely. These findings suggest that multimodal fusion acts as an implicit safeguard.

Post-fusion gradients lack the fine-grained details of the input document, indicating that future privacy-preserving strategies may focus on protecting pre-fusion layers rather than all layers indiscriminately.

## 6. Defense

In response to the threats posed by GI-DQA, we introduce *Safe Template*, a mitigation strategy that targets the adversary's dependence on publicly available document templates. Since organizations control these templates, they can proactively embed subtle, human-imperceptible perturbations into published versions to hinder reconstruction efforts. This defense is based on a key insight: gradient inversion attacks heavily depend on gradient matching, wherein the adversary aligns their reconstructions with the observed shared gradients. By maximizing gradient divergence at the input level, *Safe Template* disrupts this alignment without altering the visual appearance of the document. Unlike prior defenses that manipulate or obfuscate the gradients themselves (Li et al., 2022; Zhu et al., 2019), Safe Template perturbs the input images used to compute those gradients. We evaluate two strategies for introducing perturbations: (i) random noise, and (ii) adversarial noise crafted using projected gradient descent (PGD) (Madry et al., 2017), with a custom objective $\mathcal{L}_{\text{defender}} = -\mathcal{L}_{\text{grad}}$ that explicitly maximizes gradient divergence. Both approaches prove to be effective in reducing the success of the attack. Random noise is effective for $\epsilon > \frac{32}{255}$, while PGD-based perturbations provide comparable defense even at $\epsilon > \frac{8}{255}$, where $\epsilon$ denotes the perturbation budget, which controls the intensity of modifications applied to the images. Importantly, *Safe Template* is applied offline, imposes no computational overhead on the training process, and preserves full model utility, since benign clients train on clean, unmodified documents. This makes it a lightweight and practical defense for safeguarding multimodal FL systems. For detailed results and analysis, see Appendix A.2.

## 7. Conclusion

In this work, we introduced GI-DQA, a gradient inversion attack that reveals critical privacy vulnerabilities in multimodal federated learning (FL) systems. By exploiting prefusion gradients and cross-modal interactions, GI-DQA successfully reconstructs private document content, exposing a significant threat overlooked by prior unimodal approaches, which fail in such settings. To address this risk, we proposed *Safe Template*, an input-level defense that perturbs shared templates to disrupt gradient alignment, effectively mitigating inversion without affecting model utility. Our findings underscore the need for privacy-aware architectural design and robust defensive strategies in multimodal FL deployments. Future work should investigate adaptive defenses and training protocols that preserve both utility and privacy in cross-modal learning environments.

## Impact Statement

This paper presents a study on gradient inversion attacks in FL, specifically targeting DQA models. Our work exposes significant privacy vulnerabilities in FL systems, particularly in scenarios where sensitive document data are involved, such as medical records, financial statements, and legal contracts.

**Potential Ethical Concerns.** Federated learning, despite its privacy-preserving design, remains susceptible to gradient inversion attacks that can reconstruct private training data, posing a risk to individuals whose confidential information may be leaked. This is especially concerning for applications involving sensitive or personal documents. Althoughhugh our research aims to improve the security of FL systems by exposing these vulnerabilities, it also carries the risk of adversarial actors misusing our findings to develop more effective attacks against real-world deployments.

**Mitigation and Responsible Use.** Our work underscores the urgent need for robust defenses, such as gradient obfuscation techniques, differential privacy mechanisms, and secure aggregation strategies, to mitigate the identified risks. We also present a defense mechanism to mitigate the risks posed by such attacks. We encourage the research community to use our findings to strengthen privacy-preserving methods in FL rather than to exploit them maliciously. Furthermore, we adhere to responsible disclosure practices and have framed our research in a way that prioritizes defense-oriented contributions.

**Broader Societal Implications.** By demonstrating vulnerabilities in multimodal FL systems, we contribute to ongoing efforts to enhance the security and privacy of decentralized machine learning frameworks. Furthermore, our study informs policymakers, developers, and researchers about potential risks in privacy-sensitive AI applications, helping to shape future regulatory and technical safeguards. In conclusion, this research seeks to advance the field of machine learning by improving the understanding of privacy risks in FL, with the ultimate goal of fostering more secure and privacy-aware AI systems.

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

# A. Appendix

## A.1. Additional Results

### A.1.1. ATTACK PERFORMANCE ON BATCHES

We also examine the effect of different batch sizes on the quality of reconstruction, as it reflects a more complex scenario in which the shared gradients are averaged over the entire batch. In our experiments, we demonstrate the ability to reconstruct a single document when the shared gradients are computed using various batch sizes. Table 4 presents the results of the reconstruction using batch sizes of two, four, and six. Our experiments demonstrate that our attack remains effective when gradients are computed from up to two samples, successfully reconstructing a single document in these cases. However, with larger batch sizes, the quality of reconstruction diminishes. This effect stems from the nature of text-based gradients,

Table 4: Effect of different batch sizes on the reconstruction quality. ↓ indicates that lower values are better, and ↑ indicates that higher values are better.

| Batch Size | Reconstruction Metrics | | | | |
|---|---|---|---|---|---|
| | PSNR ↑ | FFT$_{2D}$ ↓ | MSE ↓ | Binary ↑ | Fuzz Ratio ↑ |
| 2 | 15.967 | 0.014 | 83.57 | 49.6% | 0.452 |
| 4 | 7.155 | 0.132 | 90.461 | 0.7% | 0 |
| 6 | 8.215 | 0.094 | 94.001 | 0.6% | 0 |

which tend to be more sparser and localized compared to visual features. When gradients are averaged over larger batches, the document-specific signal becomes more diffused, making it harder to isolate fine-grained textual details. Nevertheless, despite this increased complexity, our approach remains capable of extracting meaningful information, demonstrating its robustness in multi-sample scenarios.

### A.1.2. TRAINING STAGE EFFECT

As noted in Section 4.2.2, we conducted a comprehensive evaluation of the effectiveness of the attack at various stages of model training. As shown in Figure 6, the ability to reconstruct document pixels progressively degrades as the model converges. For example, at initialization (iteration 0), approximately 70.1% of words are perfectly reconstructed, whereas by iteration 30, the success rate drops below 10%.

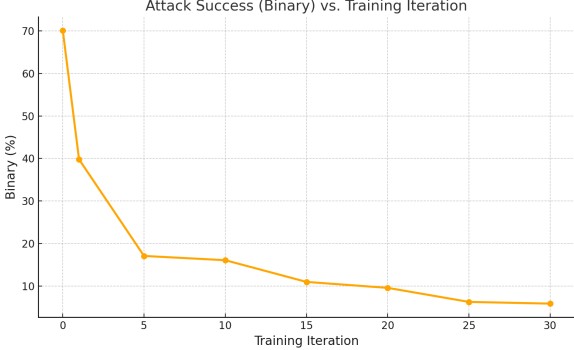

Figure 6: Attack success rate (Binary accuracy) as a function of training iteration. The results show a clear downward trend in reconstruction performance as the model converges.

### A.1.3. EFFECT OF PRIOR SCHEDULING STRATEGY

To assess the impact of the prior scheduling strategy on reconstruction quality, we compare our proposed approach with two alternatives: (i) the delayed scheduling strategy of GradViT (Hatamizadeh et al., 2022), and (ii) a baseline without any scheduling mechanism, where prior weights are kept constant throughout the optimization. GradViT delays the introduction of auxiliary priors until the later stages, allowing the model to initially optimize solely based on the gradient matching loss. In contrast, our scheduler assigns higher weights to auxiliary priors during early iterations and gradually reduces their influence via an exponential decay schedule. This design aims to stabilize the early stages of optimization, especially in settings where the template reduces the gradient signal in sensitive regions, by encouraging visually coherent structures from the outset.

Table 5 summarizes the results. Our scheduler is the only one to produce viable reconstructions, achieving strong visual and

semantic fidelity across all metrics. In contrast, both GradViT and the constant-weight baseline fail to guide the optimization meaningfully: GradViT's delayed prior activation leads to noisy and fragmented outputs, while constant weighting results in low-fidelity reconstructions that collapse semantically (*i.e.*, near-zero binary accuracy and fuzz ratio). These findings highlight the importance of strong early guidance from auxiliary priors when gradient signals are weak or ambiguous.

Table 5: Comparison of prior scheduling strategies. Our early-weighted scheduler leads to significantly improved reconstructions across visual and semantic metrics.

| Scheduler | PSNR ↑ | FFT$_{2D}$ ↓ | MSE ↓ | Binary ↑ | Fuzz Ratio↑ |
|---|---|---|---|---|---|
| None | 10.854 | 0.048 | 94.781 | 5.1% | 0 |
| GradViT | 8.871 | 0.080 | 79.440 | 0.1% | 0.042 |
| Ours | 24.194 | 0.003 | 60.403 | 70.1% | 0.909 |

### A.1.4. EFFECTIVENESS OF INDIVIDUAL PRIORS

To better understand the role of each auxiliary prior in our reconstruction objective, we evaluate them independently rather than cumulatively (Section 4.2.2). Table 6 presents the results of optimizing each loss term in isolation. The Laplacian prior ($\mathcal{R}$txt) and the spatial TV prior ($\mathcal{R}$TV-S) achieve the strongest overall performance, significantly improving both perceptual quality (highest PSNR) and textual fidelity (lowest fuzz ratio, highest binary match). The channel TV prior ($\mathcal{R}_{TV-C}$) also performs well, particularly in maintaining semantic consistency. In contrast, the Gaussian prior ($\mathcal{R}$gau) achieves relatively modest performance when used alone. However, this does not reflect its true value; when used in combination with other priors (as shown in our main ablation), it plays a critical role in stabilizing optimization and suppressing noise. Finally, attacks without any regularization ($\mathcal{L}$grad only) exhibit degraded results across all metrics, highlighting the necessity of auxiliary priors for successful reconstruction.

Table 6: Performance of individual priors based on reconstruction metrics.

| Loss Component | PSNR ↑ | FFT$_{2D}$ ↓ | MSE ↓ | Binary Test ↑ | Fuzz Ratio ↑ |
|---|---|---|---|---|---|
| $\mathcal{L}_{grad}$ | 17.872 | 0.012 | 75.008 | 42.6% | 0.759 |
| $\mathcal{R}_{gau}$ | 18.246 | 0.010 | 74.701 | 45.1% | 0.771 |
| $\mathcal{R}_{txt}$ | 21.182 | 0.005 | 71.783 | 66.6% | 0.896 |
| $\mathcal{R}_{TV-C}$ | 20.519 | 0.007 | 71.501 | 60.8% | 0.874 |
| $\mathcal{R}_{TV-S}$ | 20.508 | 0.006 | 72.915 | 64.2% | 0.888 |
| All Combined | 24.194 | 0.003 | 60.403 | 70.1% | 0.909 |

## A.2. Defense

We propose *Safe Template*, a practical defense mechanism designed to mitigate gradient inversion attacks on multimodal models, particularly DQA models within FL setups. This defense is motivated by a key vulnerability exploited in our attack: the adversary's access to publicly available document templates. These templates are frequently shared by organizations for standardization and accessibility purposes, and their consistent structure enables adversaries to significantly narrow the search space during reconstruction. The success of a gradient inversion attack depends on the similarity between gradients computed on the attacker's reference template and those generated by the actual client. *Safe Template* disrupts this similarity by embedding small, targeted perturbations into publicly released templates, thereby diminishing their usefulness for adversarial inversion.

### A.2.1. DEFENSE OVERVIEW

*Safe Template* introduces perturbations into the document template that maximize the distance between the gradients computed on clean versus perturbed inputs. These perturbations are generated offline, prior to FL training and do not interfere with client utility. The benign client continues to use the unperturbed document during FL training, ensuring model accuracy and gradient integrity.

---

**Algorithm 1** *Safe Template*

---

**Input:** Model $f_\theta$, visual document $x_D$, question $x_Q$, answer $y_{\text{ans}}$, perturbation budget $\epsilon$, norm $p$, step size $\alpha$
**Output:** Perturbed Template

1:  $\delta := 0$
    ▷ Step 1:
2:  $\hat{y}_{\text{ans}} = f_\theta(x_D, x_Q)$
3:  $\mathcal{L}_{\text{clean}} = \ell_{\text{model}}(\hat{y}_{\text{ans}}, y_{\text{ans}})$
4:  $\nabla\theta_{\text{clean}} = \frac{\partial \mathcal{L}_{\text{clean}}}{\partial \theta}$
    ▷ Step 2:
5:  **repeat**
6:      $x_D^{per} = x_D + \delta$
7:      $\hat{y}_{\text{per}} = f_\theta(x_D^{per}, x_Q)$
8:      $\mathcal{L}_{\text{per}} = \ell_{\text{model}}(\hat{y}_{\text{per}}, y_{\text{ans}})$
9:      $\nabla\theta_{\text{per}} = \frac{\partial \mathcal{L}_{\text{per}}}{\partial \theta}$
    ▷ Step 3:
10:     $\mathcal{L}_{\text{def}} = -\mathcal{L}_{\text{grad}}$ ▷ see Section 3.3.1
11:     $\nabla\theta_{\text{def}} = \frac{\partial \mathcal{L}_{\text{def}}}{\partial \theta}$
12:     $\delta = \prod_{||\delta||_p < \epsilon} (\delta + \alpha \cdot \text{sign}(\nabla\theta_{\text{def}}))$
13:  **until** Convergence
14:  **Return** $\delta$

---

The method consists of three steps, outlined in Algorithm 1, which can be summarized as follows:

- **Step 1: Clean Gradient Computation.** Given a document-question pair $(x_D, x_Q)$, the model performs a forward pass and predicts an answer $\hat{y}$. The loss is computed with respect to the ground-truth answer $y$, and the gradients $\nabla\theta_{\text{clean}}$ are obtained by backpropagation. This step is performed once per template.
- **Step 2: Adversarial Gradient Computation.** A perturbation $\delta$ is added to the document, yielding $(x_D + \delta, x_Q)$. The model performs a forward pass and backpropagation to compute $\nabla\theta_{\text{per}}$, the gradients on the perturbed template.
- **Step 3: Perturbation Update.** The perturbation $\delta$ is optimized to maximize the gradient discrepancy, using the defender's loss $\mathcal{L}_{\text{def}} = -\mathcal{L}_{\text{grad}}$. Gradients with respect to $\mathcal{L}_{\text{defender}}$ are computed, and $\delta$ is updated via Projected Gradient Descent (PGD) to remain within an $\ell_p$ norm ball of radius $\epsilon$:

$$\delta_{t+1} = \prod_{\|\delta\|_p \leq \epsilon} (\delta_t + \alpha \cdot \text{sign}(\nabla\mathcal{L}_{\text{def}})),$$

where $\alpha$ is the step size and $\prod$ denotes projection under norm constraint $\|\cdot\|_p$. Steps 2 and 3 are repeated iteratively until convergence.

### A.2.2. KEY PROPERTIES

*Safe Template* provides several practical advantages:

- **Offline and Efficient.** The perturbation generation process is performed once per template and does not interfere with FL training, making it computationally efficient.

- **Accuracy Preservation.** Since benign clients use the original, unperturbed documents during training, model accuracy and utility are fully preserved. Gradients transmitted to the server remain unaffected.

- **Scalable and Deployable.** Organizations can embed perturbations directly into the public template files they distribute, requiring no client-side or server-side modification.

### A.2.3. RANDOM VS. ADVERSARIAL PERTURBATIONS

To evaluate the robustness of SafeTemplate, we explore two strategies for perturbation generation: (i) adversarially crafted perturbations, and (ii) random noise; Both approaches modify document pixels directly in a way that remains imperceptible

to the human eye, preserving the visual integrity of the template. To investigate the trade-off between privacy and document fidelity, we experiment with varying perturbation budgets, denoted by $\epsilon$, which control the maximum allowable pixel modification. Table 7 reports the effectiveness of both strategies on the Donut model.

The results demonstrate that both methods substantially reduce the success of the attack. Random perturbations become effective at $\epsilon > \frac{32}{255}$, lowering the reconstruction success rate to just 0.3%. In contrast, adversarial perturbations achieve similar protection at a much smaller budget ($\epsilon > \frac{8}{255}$), highlighting that even minimal, strategically crafted modifications are sufficient to disrupt gradient inversion.

Table 7: $\downarrow$ indicates that lower values are better, and $\uparrow$ indicates that higher values are better. Bold indicates superior results.

| Strategy | $\epsilon$ | Reconstruction Metrics | | | | |
|----------|-----------|-----------|-----------|-----------|-----------|-----------|
| | | PSNR $\downarrow$ | FFT$_{2D}$ $\uparrow$ | MSE $\uparrow$ | Binary $\downarrow$ | Fuzz Ratio $\downarrow$ |
| No Defense | - | 24.194 | 0.003 | 60.403 | 70.1% | 0.909 |
| Random | 12/255 | 13.905 | 0.024 | 80.876 | 11.6% | 0.064 |
| | 32/255 | 9.036 | 0.073 | 84.946 | 0.3% | 0 |
| PGD | 8/255 | 12.027 | 0.037 | 84.550 | 0.25% | 0.01 |
| | 12/255 | 10.210 | 0.056 | 86.014 | 0.4% | 0.002 |
| | 16/255 | 9.004 | 0.075 | 86.961 | 0.4% | 0.001 |

### A.2.4. FL WITH LOCAL DIFFERENTIAL PRIVACY (LDP)

To assess the effectiveness of standard FL defenses, we conducted a complementary experiment simulating a typical LDP mechanism. Specifically, we added Gaussian noise directly to the shared gradients, a widely adopted strategy for per-client privacy preservation in FL. As shown in Table 8, increasing the noise scale $\sigma$ leads to a gradual reduction in attack success. At lower noise levels ($\sigma = 1 \cdot 10^{-9}$ and $1 \cdot 10^{-8}$), the gradients still retain enough signal for partial reconstruction, with binary match rates around 44–46%. At a higher noise scale ($\sigma = 1 \cdot 10^{-7}$), the attack is largely neutralized, although this level of perturbation may adversely affect model convergence and utility in practice. These findings highlight the limitations of gradient-level defenses: stronger privacy comes at the cost of model performance. In contrast, *Safe Template* operates entirely at the input level without modifying gradients, preserving training stability while substantially degrading the adversary's ability to extract sensitive information.

Table 8: Effectiveness of LDP under different noise scales. Higher $\sigma$ implies stronger noise added to shared gradients.

| $\sigma$ | PSNR $\downarrow$ | FFT$_{2D}$ $\uparrow$ | MSE $\uparrow$ | Binary $\downarrow$ | Fuzz Ratio $\downarrow$ |
|----------|-----------|-----------|-----------|-----------|-----------|
| No Defense | 24.194 | 0.003 | 60.403 | 70.1% | 0.909 |
| $1 \cdot 10^{-9}$ | 19.566 | 0.007 | 74.516 | 45.9% | 0.786 |
| $1 \cdot 10^{-8}$ | 17.805 | 0.010 | 77.412 | 44.0% | 0.825 |
| $1 \cdot 10^{-7}$ | 7.013 | 0.119 | 82.807 | 0% | 0.001 |

### A.3. Visualizations

In Figure 7, we visualize full-sized reconstructed documents where private information is recovered.

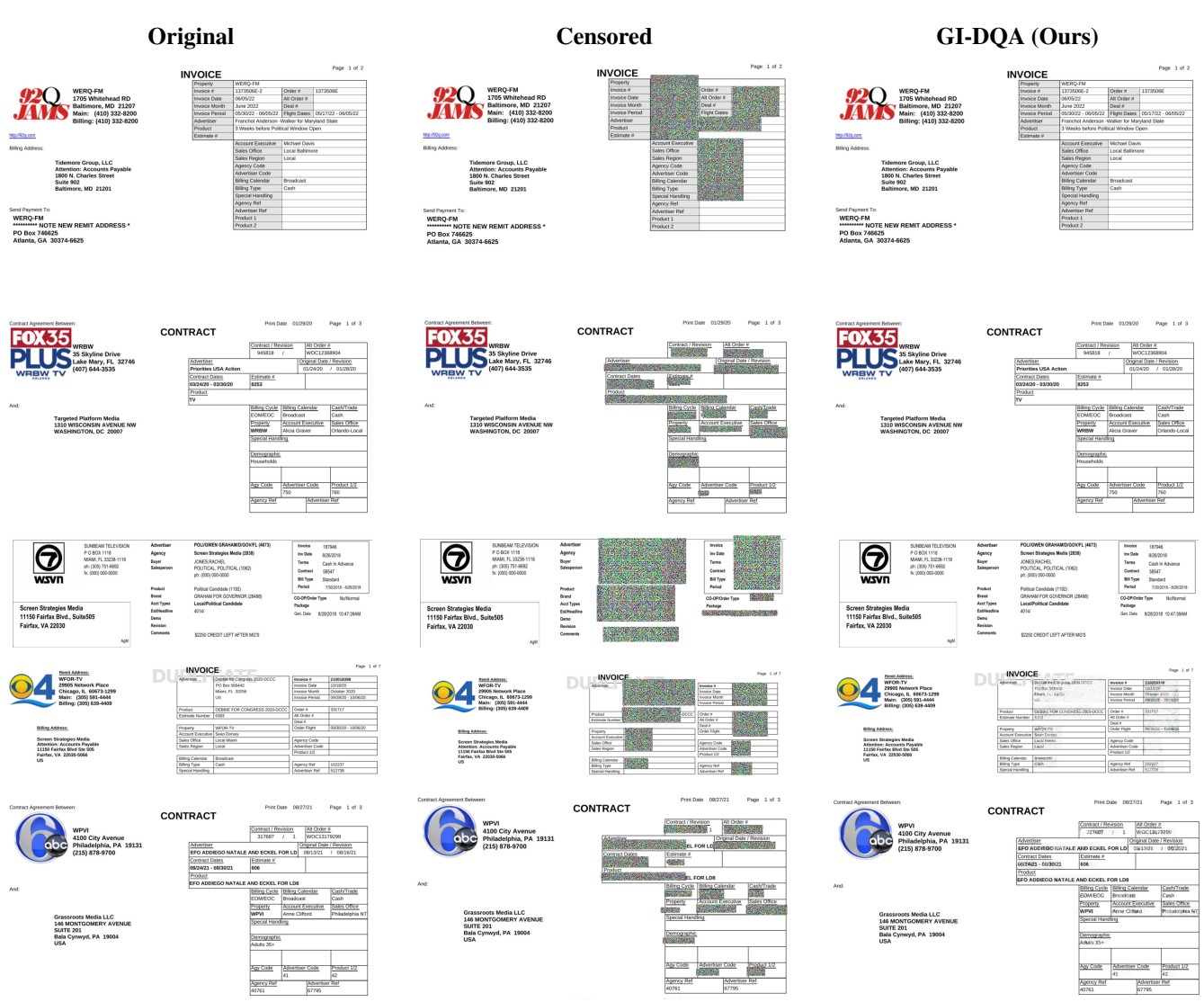

Figure 7: Examples of documents where private information is reconstructed using our proposed attack. Censored represents the document with redacted private data, which serves as the adversary's initial reference before optimization.

