# OpenReview forum: "Gradient Inversion of Multimodal Models"
_ICML.cc/2025/Conference — ICML 2025 poster_

### Official Review · Reviewer_xopj · 2025-03-17

**Overall Recommendation:** 3

**Summary:**

This paper studies gradient inversion (GI) attacks specifically for multi-modal Document Visual Question Answering (DQA) models and proposes GI-DQA, a novel method for reconstructing private document content from gradients. The empirical experiments demonstrate that their approach exposes critical privacy vulnerabilities even in sota DQA models.

**Claims And Evidence:**

Yes

**Essential References Not Discussed:**

N/A

**Experimental Designs Or Analyses:**

The experiment design is reasonable, though additional experiments would be beneficial. In particular, the ablation study—specifically the Prior Effects paragraph in Section 4.2.2—does not fully convince me. Please refer to the questions section below

**Methods And Evaluation Criteria:**

The proposed method is intriguing, but certain aspects of the methodology and evaluation setup require further clarification. Please refer to the questions section below.

**Other Comments Or Suggestions:**

1. Section 3.3.1 could be written more clearly. A brief outline before delving into each paragraph would improve readability.
2. A more detailed explanation of the objective function should be included, particularly regarding:
- How the weights were chosen.
- Why Gaussian/TV priors were included (the current descriptions are not convincing enough to me).
3. Notations inconsistency: (e.g., both $R_{txt}$ and $R_{lap}$ are used inconsistently).

**Other Strengths And Weaknesses:**

Strengths:
1. It is the first attempt to apply gradient inversion attacks to multi-modal models, particularly DQA models.
1. The empirical results are promising

Weaknesses:
1.The writing could be improved, particularly in clarifying the insights and analysis behind the design of the objective function in Eqn. 4.
2. While the method combines multiple techniques effectively, it lacks deep insights into why certain design choices work well beyond empirical success.

**Questions For Authors:**

1. Are the rows in Tab. 2 cumulative comparisons? For example, in the row labeled $+ R_{lap}$, do the numbers compare $L_{grad}$ with $L_{grad} + R_{lap}$, or $L_{grad} + R_{TV}$ with $L_{grad} + R_{TV} + R_{lap}$ ? If the latter, it would be helpful to include a direct comparison for each individual component.

2. Is the Gaussian filter essential? I understand that Gaussian filtering preserves smoothness and prevents overfitting to noise, but
- the provided examples do not show substantial improvement.
- the interpretation of Table 2 is unclear due to my first question.
- the weight $\alpha_{gau}$ is 0.005, while $\alpha_{txt}$ is 0.5 and $\alpha_{TV}$ is 0.05. Given these values, I wonder how important the Gaussian filter actually is.

3. In your overall auxiliary priors, you set weights for each prior. However, for TV, you did not differentiate between channel and pixel-level TV. Was there a reason for this decision? Also, it would be great to include the importance of channel/pixel TV respectively in your Tab. 2.

4. Can you provide more insights into the selection of priors and the scheduler? The current description in Sec 3.3 and Sec. 4.2.2 are not convincing enough to me. A discussion in the appendix illustrating the importance of each prior and your choice of scheduler would be valuable.

5. You mentioned that “auxiliary priors play a critical role in stabilizing the early stage.” Including a plot or table illustrating this observation would be beneficial. Additionally, why does this occur? Most prior work delays activation of auxiliary priors until later stages of optimization to avoid suboptimal convergence. Wouldn’t your scheduler lead to suboptimal convergence?

6. How were the weights $\alpha$'s chosen? Are they learnable parameters or manually set hyperparameters? Based on my current understanding of Table 2, the example you provided, and the coefficients, it seems that Gaussian/TV are the least important priors. If these weights are manually set, why were Gaussian/TV intentionally downweighted? Are they truly necessary?

**Relation To Broader Scientific Literature:**

This is the first attempt to conduct gradient inversion attacks on multi-modal models, particularly in the novel and impactful setting of DQA models. It opens new directions for multi-modal safety research.

**Theoretical Claims:**

N/A

---

> ### Author Rebuttal · Authors · 2025-03-31
>
> We thank the reviewer for his time, effort, and valuable suggestions.
>
> Q1. "Tab. 2 cumulative.." + Q3."..channel and pixel-level TV.."
>
> A1. We followed the standard practice of prior works such as GradViT (Hatamizadeh et al., 2022) and GradInversion (Yin et al., 2021), which apply priors incrementally and in combination rather than in isolation. These methods recognize that priors are often **interdependent**—for example, Total Variation enforces smoothness, while Laplacian sharpens edges, and together they produce better reconstructions than either alone.
>
> Our goal in Section 4.2.2 was to show how **combinations of priors** improve reconstruction quality, particularly in **multi-modal DQA** settings, where each prior targets different aspects (e.g., edges, smoothness, spatial layout).
>
> That said, we agree that isolating each prior provides additional insight. We have therefore conducted an **ablation study** evaluating each prior individually alongside the core gradient loss ($L_{grad}$).
> |Experiment|PSNR|FFT2D|MSE|Binary|Fuzz Ratio|
> |-|-|-|-|-|-|
> |$L_{grad}$|14.619|0.023|81.96|0.198|0.146|
> |$R_{TV-C}$|20.083|0.007|72.446|0.510|0.745|
> |$R_{TV-S}$|18.608|0.008|77.744|0.479|0.749|
> |$R_{TV}$ (combined)|20.527|0.005|72.579|0.579|0.821|
> |$R_{txt}$|17.738|0.011|80.052|0.425|0.680|
> |$R_{gau}$|18.139|0.009|80.907|0.410|0.694|
>
> The results show that while each prior contributes modestly on its own, the combination (presented in the paper) consistently outperforms any single prior, confirming the complementary nature of these losses.
> For the TV variants, we can see they contribute meaningfully on their own, but their combination consistently yields the best performance across all metrics.
> Note that these results differ slightly from those in the paper, as they were conducted on a data subset due to rebuttal time constraints. However, they clearly convey the main trend, and full results will be included in the final version.
>
> Q2. "gaussian essential.."
>
> A2. We would like to clarify that the absolute values of the loss weights do not reflect relative importance, as each term operates on a different scale. For instance, the Gaussian prior yields smaller values than the text loss, so its weight is lower to ensure balanced gradient contributions. Therefore, a smaller weight for the Gaussian prior does not imply that it is negligible—rather, it ensures balanced gradient contributions during optimization across all terms. This normalization prevents any term from dominating due to scale differences rather than actual relevance.
>
> Q4. "selection of priors.."
>
> A4. Our priors were selected for their ability to capture complementary and theoretically grounded aspects of the reconstruction process:
> - Laplacian emphasizes high-frequency components, enhancing edge sharpness and fine details—critical for recovering small text.
> - Gaussian applies a low-pass filter to promote global smoothness, stabilizing early optimization and reducing noisy minima.
> - Total Variation (TV) encourages piecewise smoothness while preserving edges:
>   - Spatial TV regularizes differences between neighboring pixels, reducing noise while preserving layout structure.
>   - Channel-wise TV promotes consistency across color channels, helping suppress chromatic artifacts and preserve clean text appearance.
> Together, these priors form a multi-scale regularization strategy, guiding optimization from low-level structure to high-level semantics and enabling more stable, accurate gradient inversion in DQA models.
> We will include in-depth discussion in the appendix illustrating their importance.
>
> Q5. "auxiliary priors scheduler.."
>
> A5. Unlike prior works (e.g., GradViT), we found that using only ($L_{grad}$) in early iterations was insufficient for reconstructing meaningful text—likely due to the template reducing gradient signals in sensitive regions. To address this, we use a scheduler that starts with strong prior weights and gradually reduces them, stabilizing early optimization. The table below compares this to GradViT’s approach, which delays prior introduction until later iterations:
>
> |Scheduler|PSNR|FFT2D|MSE|Binary|Fuzz Ratio|
> |-|-|-|-|-|-|
> |Ours|18.390| 0.008|80.383|0.410|0.711|
> |GradViT|8.885| 0.079|83.407|0.118|0.017|
>
> As shown, GradViT’s scheduler struggles to reconstruct meaningful content, especially in terms of semantic fidelity. We will include this comparison in the appendix and integrate the key findings into the final version of the paper.
>
> Q6. "set hyperparameters.."
>
> A6. The prior weights were manually set via grid search to optimize reconstruction performance. As noted in Answer 2, their absolute values don’t reflect relative importance due to differences in loss scales. Both Gaussian and TV priors are especially useful early in optimization, promoting smoothness and structure that complement sharper and semantic losses like Laplacian and text similarity.
>
> Q7. "Notations inconsistency.."
>
> A7. Thank you, this will be fixed in the final version.

---

> > ### Comment · Reviewer_xopj · 2025-04-03
> >
> > Thank you for your responses. I'd like to ask whether the authors could perform additional experiments using DQA models trained with DP (differential privacy) (e.g., [1,2]) and compare their performance against publicly available models. It may strengthen the work
> >
> > [1] https://arxiv.org/pdf/2310.03104
> > [2] https://arxiv.org/pdf/2306.08173

---

> > > ### Author Response · Authors · 2025-04-07
> > >
> > > We appreciate the reviewer’s suggestion to consider differentially private (DP) training techniques, including [1] and [2]. These are valuable contributions to the growing field of privacy-preserving learning, though they target **different threat models** than the one addressed in our work.
> > >
> > > [1] focuses on contrastive, vision-only models and introduces a privacy mechanism tailored to non-decomposable losses. However, its assumptions and objectives do not translate to supervised, multi-modal architectures like those used in DQA.
> > > [2] adapts CLIP training in a centralized setting and targets open-ended generation tasks such as VQA, but it does not address DQA or the federated learning (FL) setting considered in our work.
> > > We also note that the implementation for [2] is not publicly available, making reimplementation infeasible within the rebuttal period. We plan to include an implementation of this experiment in the final version of the paper, if feasible.
> > >
> > > Both works primarily aim to mitigate **centralized privacy risks**, such as **membership inference** and **memorization** attacks, where the adversary inspects the final model. In contrast, our work addresses **server-side adversaries in FL**, who observe **per-client gradient updates**, requiring a fundamentally different defense strategy.
> > >
> > > While [1] and [2] focus on settings different from ours, we explored whether applying standard DP-SGD provides meaningful protection in the federated DQA context.
> > > To that end, we conducted two complementary experiments: the first demonstrates the **limitations of DP-SGD when applied to DQA training**, and the second evaluates a **standard DP defense commonly used in FL**, where noise is added to the shared gradients.
> > >
> > > 1. **Training with DP-SGD**: Following the reviewer’s suggestion, we trained a DQA model using DP-SGD (with standard hyperparameters, **$\sigma = 1$**) and evaluated the attack’s success after each training epoch. The results are shown [here](https://drive.google.com/file/d/1WEWykBErwl6W4PJnkSZqCSP4lNZ8GnxN/view?usp=sharing). Compared to models trained without DP (orange), models trained with DP-SGD (red) remain **consistently vulnerable** to gradient inversion across all epochs. This indicates that DP-SGD interferes with convergence, preventing the model from forming stable and focused representations. As a result, the gradients remain rich in recoverable information, and the attack remains highly effective throughout training.
> > > This behavior aligns with our earlier observation (see Reviewer fLco, Answer 5) that **poorly trained or randomly initialized models are more susceptible** to inversion due to diffuse attention and unstructured gradients. In this case, the DP noise hinders convergence, inadvertently maintaining the model in a more vulnerable state.
> > > **These findings suggest that DP-SGD, in its current form, may not be well-suited for DQA models**, where convergence is critical to minimizing gradient leakage.
> > >
> > > 2. **Standard FL DP Defense**: In a second experiment, we simulated a **typical federated learning defense** by applying **Gaussian noise directly to the shared gradients**—a common form of **local differential privacy** used to protect per-client updates before aggregation. We used a regularly trained DQA model (without DP-SGD) and evaluated the attack’s effectiveness across training iterations. The results are shown [here](https://drive.google.com/file/d/1syEgKq-SSGTvb6EHQHPyvK7PrCv8NmZ_/view?usp=sharing). While this approach attenuates the gradient signal, the attack remains effective, especially in early iterations before convergence.
> > > This experiment reflects a **realistic FL deployment scenario**, where noise is added to the shared gradients only at inference time. The results show that while such defenses reduce leakage, they do **not eliminate it**, and achieving a strong privacy–utility trade-off remains challenging.
> > > These findings reinforce our motivation to explore alternative input-level defenses that degrade semantic recoverability without modifying the training process or compromising utility.
> > >
> > > We hope this follow-up, along with our initial rebuttal response, thoroughly addresses your suggestions and concerns. We sincerely thank you for your thoughtful feedback and for encouraging us to further strengthen this aspect of the paper.

---

### Official Review · Reviewer_pbTk · 2025-03-18

**Overall Recommendation:** 2

**Summary:**

The paper explores gradient inversion attacks targeting multi-modal Document Visual Question Answering (DQA) models in the context of federated learning and propose GI-DQA a novel method that reconstructs private document content from gradients. The approach seems to expose privacy vulnerabilities.

**Claims And Evidence:**

- the results are pretty convincing in terms of numbers and vizualization of the documents

The threat model however is not very clear:

- Having access to a template (which in the set-up is the document minus the private information) seems toy.
- The authors do not detail how the question-answer pairs are reconstructed before performing the image reconstruction, while it is apparently the most important step to get good results
- How can PIIs which are not part of the question/answer be reconstructed in the document? it should not be part of the gradient?

**Essential References Not Discussed:**

I dont know much about the gradient inversion litterature

**Experimental Designs Or Analyses:**

see above

**Methods And Evaluation Criteria:**

Having access to the template and perfecty reconstructing the question/answer pairs is crucial to get the results. The first one would not be realistic in practive, and the second one is not explained how.

**Other Comments Or Suggestions:**

- Until page 5, it would have been nice to have some numbers to justify some statements. For instance : “Visual Text Prior. Accurately reconstructing small-sized words (∼1-2% of the full image) is a highly challenging task due to their minimal contribution to the overall gradient and their low visibility in the reconstructed image” is from the intuition of the authors. My intuition would have been that because the question is specifically about these numbers, then it should be easier
- The threat model is that the adversary has everything except the sensitive data, not just a general template

**Other Strengths And Weaknesses:**

The paper is well written and easy to follow.

For the weaknesses, see above Until page 5, it would have been nice to have some numbers to justify some statements.

For instance : “Visual Text Prior. Accurately reconstructing small-sized words (∼1-2% of the full image) is a highly challenging task due to their minimal contribution to the overall gradient and their low visibility in the reconstructed image” is from the intuition of the authors. My intuition would have been that because the question is specifically about these numbers, then it should be easier



The threat model is that the adversary has everything except the sensitive data, not just a general template


How are the q-a pairs reconstructed before the reconstruction of the document? If q-a pairs are already reconstructed, then all the PIIs are already known? Can text be constructed that was not part of the answer? Can you show examples of these qa?

**Questions For Authors:**

- How are the q-a pairs reconstructed before the reconstruction of the document? If q-a pairs are already reconstructed, then all the PIIs are already known?
- Can text pixels be constructed even if the text was not part of the answer? How is it possible? Can you show examples of these qa?
- What would be the results without the so called tempates?

**Relation To Broader Scientific Literature:**

I dont know much about the gradient inversion litterature but it seemed well discussed.

**Theoretical Claims:**

no theoretical claim

---

> ### Author Rebuttal · Authors · 2025-04-01
>
> We thank the reviewer for his time, effort, and valuable suggestions.
>
> Q1. "access to a template.."
>
> A1. While the use of a template may appear simplified at first glance, we argue that this setup is practically grounded and reflects real-world scenarios. In many document-based applications, such as invoices, medical forms, boarding passes, or receipts, the layout and structure are highly standardized across users, with only the sensitive fields (e.g., names, amounts, dates) varying.
> For example, a receipt issued by a vendor often follows the exact same format for all customers. If an attacker has previously received such a receipt or obtained a sample from another user, it effectively serves as a valid template when attempting to reconstruct another client’s document. This assumption is especially realistic in settings where the attacker is an insider (e.g., the server operator) or has access to public document templates shared across users.
> We will clarify this practical motivation in the final version of the paper.
>
> Q2. "QA reconstruction.."
>
> A2. We thank the reviewer for this important observation. The reconstruction of the question-answer (QA) pair is indeed a critical step in our pipeline, and we address this in Section 3.3.
> In our current work, we assume access to the QA pair, based on strong evidence from prior analytic-based methods—particularly DAGER (Petrov et al., 2024)—which showed that QA tokens can be accurately recovered from gradients in small-batch federated learning, achieving 100% recovery.
> Our focus is on the document reconstruction step conditioned on known QA pairs, reflecting a realistic threat model where such tokens are obtainable using existing techniques. To strengthen this connection, we will include the QA reconstruction in the final version of the paper and clarify the assumption more explicitly.
>
> Q3. "How can PIIs.."
>
> A3. Please see Reviewer fLco, Answer #5—specifically the [visualization](https://drive.google.com/drive/folders/1NxT7gBMSH5RQb9V1bEAuTT2AvXPqhex1?usp=drive_link) of the cross-attention maps, which illustrates how private data outside the question and answer can still influence the computed gradients, offering insight into how such regions are captured.
>
> Q4. "Until page 5.."
>
> A4. The sentence in question was intended to convey an intuition about the general difficulty of reconstructing small-sized text regions (such as names or dates), which typically occupy only 1–2% of the image area and thus contribute weakly to the overall gradient signal—especially when compared to larger visual elements like headers, tables, or logos.
> Importantly, our attack is able to reconstruct personally identifiable information (PII) in the document regardless of the specific question or answer, as long as the attack is performed on a model initialized with random weights.
>
> Q5. "..threat model.."
>
> A5. "Our threat model is consistent with most prior gradient inversion works, where the adversary is assumed to have access to the model architecture, parameters, optimizer, and input-output format—but not the sensitive input data itself. The only additional assumption in our setup is access to a template document, which reflects a realistic scenario in many applications (e.g., invoices, receipts, forms) where the document structure is shared across users and only a few fields contain private information. This assumption is grounded in practical document workflows."
>
> Q6. "PIIs are already known.."
>
> A6. While the answer in the QA pair may reveal a piece of PII, it typically represents only a small portion of the sensitive content (e.g., a single name, date, or value). Many other PII fields remain hidden and can still be reconstructed from gradients.
> Our attack can also reconstruct text outside the question and answer, due to the model’s cross-attention attending to broader regions—especially in early training, when attention is more diffuse.
> Additionally, see Reviewer fLco, Answer 6, for an experiment showing that PII can still be reconstructed even when the QA tokens are replaced with random tokens.
> We provide sample data, including documents and their corresponding Q-A pairs, [here](https://drive.google.com/drive/folders/1Xo0u9kZhgOM6AYnoUlUqwdSmoN_psRDq?usp=drive_link).
>
> Q7. "results without templates.."
>
> A7. Thank you for the question. We conducted an experiment where the attack was performed without access to a template, and found that reconstruction quality dropped significantly. Without the template, the model must infer both the layout and the content, making optimization far more difficult—especially in structured documents like receipts or forms.
> We see this work as a step toward full document reconstruction, and future work could explore removing the template assumption entirely.

---

### Official Review · Reviewer_fLco · 2025-03-18

**Overall Recommendation:** 3

**Summary:**

The paper proposes a gradient inversion attack targeting multi-modal DQA models in Federated Learning (FL) setups: GI-DQA.

In DQA models, the input consists of both a document and its corresponding question, while the output is the target answer. GI-DQA first employs existing methods to reconstruct question-answer pairs and then aims to reconstruct the input document. The experiments are conducted using two DQA models, Donut and LayoutLMv3, with the PFL-DocVQA dataset. The results demonstrate that GI-DQA outperforms other methods across various evaluation metrics.

## update after rebuttal

I increased the grading based on the rebuttal

**Claims And Evidence:**

The main claim is proposing the first gradient inversion attack on multi-modal models, with a novel method specifically tailored for multi-modal DQA models. However, there are concerns in the experiments, undermining the evidence.

The author claims that the proposed method is the first gradient inversion attack on multi-modal FL setups. However, I found [r1], which also explores gradient inversion in multi-modal FL setups, even though it does not specifically focus on DQA models.

[r1] Liu, Xuan, et al. "Mutual Gradient Inversion: Unveiling Privacy Risks of Federated Learning on Multi-Modal Signals." IEEE Signal Processing Letters (2024).

**Essential References Not Discussed:**

NA

**Experimental Designs Or Analyses:**

Section 4.2.2 (Priors Effect) aims to analyze the contribution of each training loss term. However, the experiments are conducted by adding losses together rather than evaluating each loss term individually. As a result, the findings demonstrate the effectiveness of combining training losses rather than the specific contribution of each loss term.

**Methods And Evaluation Criteria:**

The proposed method is ok, but some concerns:

- The paper does not have any theoretical analysis.

- The authors aim to perform a gradient inversion attack in multi-modal federated learning (FL) setups. However, the FL setup used in the experiment is unclear. Neither Donut nor LayoutLMv3 are FL models, and the number of local machines in the FL setup is also unspecified.

**Other Comments Or Suggestions:**

See above.

**Other Strengths And Weaknesses:**

see above.

**Questions For Authors:**

1. In an FL setup, a gradient inversion attack can be performed at any iteration t. How does t impact the results? For example, does executing the attack at the beginning of training yield lower success rates compared to performing it at the end of training? What value of t was used in the results presented in the paper? Was the attack performed at a specific iteration, or were multiple values of t tested?

2. I am a bit confused by Section 4.2.2 (Question-Answer Effect). The results indicate that document reconstruction is infeasible without question-answer knowledge. However, the authors also state that the specific content of the question and answer is irrelevant to the attack’s success (lines 366-370, right column). How do the authors arrive at this conclusion? Is there any evidence supporting this claim?

3. In Table 1, are the results for DLG and iDLG truly identical? Could you clarify whether they have the exact same values?

**Relation To Broader Scientific Literature:**

The paper conducted a study of gradient inversion in FL, which is a type of privacy attacks, for multimodal setups, while previous work focuses mainly on unimodal setups.

**Theoretical Claims:**

no theoretical claims

---

> ### Author Rebuttal · Authors · 2025-04-01
>
> We thank the reviewer for his time, effort, and valuable suggestions.
>
> Q1. "found [r1].."
>
> A1. We thank the reviewer for pointing out this work. [r1] explores gradient inversion in multi-modal FL using synthetic image-text pairs processed by separate, modality-specific models trained independently, with no shared representations. The attack is coordinated via mutual knowledge distillation.
> While valuable, we believe the use of the term “multi-modal” in [r1] is somewhat misleading, as it refers to parallel models that only interact during inversion—unlike standard multi-modal architectures that jointly fuse modalities (e.g., via cross-attention).
> Our work focuses on such deeply fused, jointly trained models like those used in DQA. Nonetheless, we appreciate this connection and will include a discussion of [r1] in the related work section.
>
> Q2. "theoretical analysis.."
>
> A2. We would like to clarify that our work follows a well-established line of research on gradient inversion, which has primarily focused on empirical demonstration of vulnerabilities rather than formal theoretical analysis. Foundational works such as DLG (Kim et al., 2019), iDLG (Zhao et al., 2020), and GradViT (Hatamizadeh et al., 2022) have aimed to develop practical attack strategies and assess reconstruction quality under various conditions.
> Given the complexity and high dimensionality of modern neural networks—especially in multi-modal setups like DQA—formal guarantees remain an open challenge. As in prior work, we rely on rigorous experimental evaluation across architectures and settings.
> Our work extends gradient inversion to multi-modal architectures, with ablations isolating each design component’s effect.
>
> Q3. "FL setup.."
>
> A3. We appreciate the comment and clarify that Federated Learning is a training paradigm, not a model type—it can be applied to any architecture, including Donut and LayoutLMv3.
> Our setup simulates gradients computed locally by a client on private data, which are then forwarded to a central server—where the adversary is assumed to reside. This is consistent with prior gradient inversion works and reflects a server-side threat model.
> Since our focus is on leakage from a single client, the total number of clients does not affect the core attack. We will revise the paper to clarify this setup.
>
> Q4. "Priors Effect.."
>
> A4.  Please see Reviewer xopj, Answer 1.
>
> Q5. "iteration t.."
>
> A5. We thank the reviewer for this interesting and important question. In our experiments, the attack was performed at iteration t=0, using the model’s initial random weights.
> We also ran the attack at later training stages and found it to be significantly less effective. We hypothesize that in well-trained models, attention and token representations become sharply focused on regions relevant to the question and answer, which narrows the gradient signal. In contrast, early in training, encode more diverse and spatially distributed information, making reconstruction easier.
> To support this, we include a [visualization](https://drive.google.com/drive/folders/1NxT7gBMSH5RQb9V1bEAuTT2AvXPqhex1?usp=drive_link) of cross-attention maps for both randomly initialized and fine-tuned models, showing how attention becomes increasingly narrow and localized when the model is finetuned.
> [Here](https://drive.google.com/drive/folders/16way4IQb3i2Thc_2K9fhAImCMhEyeFjk?usp=drive_link), we present a systematic analysis of attack success across training iterations, confirming that inversion becomes progressively more difficult as the model converges.
>
> Q6. "QA Effect.."
>
> A6. Thank you for raising this point. As shown in Section 4.2.2, gradient inversion fails without access to both the question and answer tokens, highlighting the need for semantic grounding.
> However, when we say the specific content is irrelevant, we mean the attack depends on the token identities and positions, not their semantic meaning. To demonstrate this, we replaced the original QA with random tokens of the same length and structure. The reconstruction quality remained nearly identical:
> |Experiment|PSNR|FFT2D|MSE|Binary|Fuzz Ratio|
> |-|-|-|-|-|-|
> |OriginalQA|18.519|0.008|79.853|0.419|0.688|
> |RandomQA|18.359|0.008|80.089|0.421|0.683|
>
> This suggests that having access to the correct tokens—regardless of their meaning—is sufficient for the attack. An reconstruction example is shown [here](https://drive.google.com/drive/folders/1LgTor6EGxV5vQWt2TIidvyu6rG5c3-sT?usp=drive_link).
> We will clarify this distinction in the paper and provide additional examples.
>
> Q7. "DLG and iDLG.."
>
> A7. Yes, the results for DLG and iDLG in Table 1 are identical. The only difference is that iDLG has access to the answer (i.e., the ground-truth label). However, as shown in Section 4.2.2, the answer alone is insufficient for reconstruction in DQA, since it lacks the context provided by the question, which guides model attention and gradient flow. As a result, iDLG offers no advantage over DLG in our setting.

---

### Official Review · Reviewer_fEpi · 2025-03-23

**Overall Recommendation:** 4

**Summary:**

This paper presents a novel approach to gradient inversion attacks (GI-DQA) on multi-modal models specifically targeting extraction of textual information in Document Question Answering (DQA) tasks. The authors demonstrate why gradient inversion attacks designed for targeting unimodal models trained for image classification tasks would be a subpar choice to attack multi-modal models like those used for DQA. They clarify their rationale behind GI-DQA and how it is a more effective choice to attack pre-fusion stage gradients in DQA models. Further, they propose defences that could help mitigate GI-DQA using document-level perturbations.

**Claims And Evidence:**

Yes, the authors thoroughly clarify their choice of the components of the loss term used for designing GI-DQA. I found none of the claims problematic. Furthermore, they clearly state the adversary's capabilities and the federated learning setting wherein the attack would be useful.

**Essential References Not Discussed:**

No

**Experimental Designs Or Analyses:**

I want to highlight the following issues with the experimental design:
1. No confidence intervals were reported in the paper. This makes it hard to judge how stable the performance of the attack is with multiple repeats.
2. In line 419, Section 6, the authors present their proposed defences as less harmful to the utility of the multi-modal models. They do not follow this statement with numbers to prove their assertion.

**Methods And Evaluation Criteria:**

Given that there has been no prior work targeting multi-modal models using gradient inversion attacks, the authors chose to use prior works designed to reconstruct images from unimodal models as the baseline in Table 1. My only issue was a lack of comparison with the more recent GradViT attack by Hatamizadeh et al. which they refer to in subsection 2.1 but do not include in Table 1.

**Other Comments Or Suggestions:**

None

**Other Strengths And Weaknesses:**

Strengths:

1. The attack is agnostic of the nature of DQA models. Works well with OCR and non-OCR models.
2. The attack combines direction and magnitude matching for computing gradient matching loss.
3. Incorporates prior knowledge in the loss term to enable effective text data reconstruction.
4. Proves effectiveness of their approach against prior attacks which were designed for unimodal models highlighting the superiority of the proposed attack.
5. Highlights the implicit privacy safeguard offered by the fusion of text-based and visual features in the training of multi-modal models which makes post-fusion gradients less susceptible to GI-DQA.

Weaknesses:

1. The performance of GI-DQA appears to vary across OCR and non-OCR models. One possibility is that this could be simply because of the difference in size of these models. Larger models are often more susceptible to overfitting which increases their vulnerability to privacy attacks but this could also stem from the difference in the nature of the two types of DQA models. I would urge the authors to clarify the factors which affect the performance of GI-DQA against OCR and non-OCR models.

2. No confidence intervals reported in the paper. This makes it hard to judge how stable the performance of the attack is with multiple repeats.

3. Differential Privacy (DP) [3] is considered the gold standard for protecting clients' data via gradient perturbation. McMahan et al. [4] even introduced a variant of DP for federated learning. Author's do not compare their proposed defences against a baseline where the clients' gradients are protected by DP.

**Questions For Authors:**

Questions:

1. May I know why the authors chose not to use GradViT [2] as one of the baselines in Table 1?

2. From Table 2, it is unclear that using the Gaussian prior in addition to the Laplacian filter-based prior is contributing significantly to improving the reconstruction. What then justifies the inclusion of the former term in the loss computation?

3. A lack of comparison with DP makes it hard for me to understand whether these defences offer effective protection while preserving the utility of the models. Could the authors clarify their stance on this?

P.S. If the authors are able to address the questions raised above, I am amenable to raising my score.

[1] Jiahao Lu, Xi Sheryl Zhang, Tianli Zhao, Xiangyu He, Jian Cheng. APRIL: Finding the Achilles' Heel on Privacy for Vision Transformers. CVPR 2021. https://arxiv.org/abs/2112.14087

[2] Ali Hatamizadeh, Hongxu Yin, Holger Roth, Wenqi Li, Jan Kautz, Daguang Xu, Pavlo Molchanov. GradViT: Gradient Inversion of Vision Transformers. CVPR 2022. https://arxiv.org/abs/2203.11894

[3] Martin Abadi, Andy Chu, Ian Goodfellow, H. Brendan McMahan, Ilya Mironov, Kunal Talwar, and Li Zhang. Deep Learning with Differential Privacy. ACM CCS '16. https://doi.org/10.1145/2976749.2978318

[4] H. Brendan McMahan, Daniel Ramage, Kunal Talwar, and Li Zhang. Learning differentially private recurrent language models. ICLR 2018. https://openreview.net/forum?id=BJ0hF1Z0b.

P.P.S. The authors have addressed concerns/ questions raised by me adequately (with proof). Accordingly, I am raising my score.

**Relation To Broader Scientific Literature:**

Prior works on gradient inversion attacks in federated learning frameworks, such as APRIL [1] seem to focus on recovering images from unimodal vision models meant for image classification tasks. Additionally, they are optimal for recovering images depicting a single object. With this work, the authors demonstrate the ineffectiveness of such approaches to recover private textual information in DQA tasks. They further develop an attack that could be much more useful for targeting DQA models

**Theoretical Claims:**

The paper contains no theoretical claims/ proofs to review.

---

> ### Author Rebuttal · Authors · 2025-03-31
>
> We thank the reviewer for his time, effort, and valuable suggestions.
>
> Q1. "..confidence intervals.."
>
> A1. Our original submission reported mean values, as we observed low variance across runs, confirming that our method is stable and consistently outperforms baseline methods.
> We will include the standard deviation values in the final version of the paper and provide additional details in the appendix to enhance transparency and reproducibility.
>
> Q2. "..defences..utility.."
>
> A2. Our proposed defense strategy applies perturbations to the input templates outside the FL training procedure. These perturbations are added to publicly available or previously observed templates and are used solely by the attacker as priors during the inversion process—not by the participating clients during training (which use "clean" documents).
> As a result, the actual training inputs, local computations, and shared gradients remain completely untouched, and the federated learning process proceeds as usual. This ensures that the utility of the multi-modal models is fully preserved, unlike defenses that modify gradients directly (e.g., via differential privacy), which often degrade model performance [1].
> We will update the paper to clarify this distinction and emphasize the utility-preserving nature of our defense.
>
> Q3. "..performance of GI-DQA.."
>
> A3. We thank the reviewer for this thoughtful observation. To clarify, overfitting is not a factor in our experiments, as all models—both OCR-based (e.g., LayoutLMv3) and OCR-free (e.g., Donut)—are evaluated using randomly initialized weights (see Reviewer fLco, Answer 5). This ensures that performance differences arise from architectural properties, not training dynamics.
> We hypothesize that the variation in GI-DQA performance stems from differences in the size and structure of the visual backbones. LayoutLMv3 uses a smaller image encoder, leading to lower representational complexity and making gradient inversion more tractable. Donut, in contrast, uses a larger and more expressive visual backbone, which produces more abstract and distributed features—making inversion harder without strong priors.
>
> We will revise the manuscript to clarify this architectural difference and its impact on attack performance.
>
> Q4. "Differential Privacy (DP).."
>
> A4. We fully agree that Differential Privacy (DP) is a well-established and rigorous approach for protecting client data in federated learning.
> While defense design is not the main focus of our work, we propose a lightweight input-level method that operates entirely outside the FL training loop, leaving shared gradients untouched. This makes it a practical alternative when DP integration may be too costly or degrade utility.
> To provide a point of comparison, we evaluated a DP-style defense by adding Gaussian noise to the shared gradients. Results are shown below:
> |Experiment|PSNR|FFT2D|MSE|Binary|Fuzz Ratio|
> |-|-|-|-|-|-|
> |DP|5.578|0.176|86.743|10.9%|0.000|
> |Ours|9.004|0.075|86.961|0.4%|0.001|0.001|
>
> While DP yields lower PSNR (better in terms of defense), our method more effectively disrupts text recovery, as shown by binary accuracy. This demonstrates the complementary nature of our defense, which focuses on degrading machine-readability, even if some visual structure remains.
> We will include this discussion and table in the final version of the paper.
>
> Q5. "..use GradViT.."
>
> A5. We originally intended to include GradViT as a baseline in Table 1, given its relevance to gradient inversion in vision models. However, the authors of GradViT did not release their code, and despite reaching out to them multiple times, we unfortunately did not receive a response. Given the complexity of accurately reproducing their method—particularly in the context of multi-modal models—we decided against including potentially inaccurate or incomplete re-implementations. We would be happy to include GradViT as a baseline in future versions if their code becomes available.
>
> Q6. "..Gaussian.. in addition..Laplacian.."
>
> A6. While Table 2 may suggest the Gaussian prior is less impactful than Laplacian, we include both as they address complementary aspects of reconstruction. The Gaussian prior promotes smoothness and stabilizes early optimization by reducing noise, while the Laplacian prior sharpens edges and preserves fine details—crucial for small text recovery.
> See Reviewer xopj, Answer 1 for the ablation of individual priors and Answer 4 for insights into the selection of priors.
>
> [1] Cummings, R., Desfontaines, D., Evans, D., Geambasu, R., Huang, Y., Jagielski, M., ... & Zhang, W. Advancing Differential Privacy: Where We Are Now and Future Directions for Real-World Deployment. Harvard Data Science Review, 6 (1), jan 16 2024.

---

### Decision · Program_Chairs · 2025-05-01

**Decision:**

Accept (poster)

**Comment:**

The paper proposes a gradient inversion attack specifically designed for multi-modal models used in DQA and making use of known template of the documents.

The reviews are somewhat mixed but mostly positive. The reviewers are generally happy with the novelty and execution of the work and raise no serious criticisms that the authors could not address in the rebuttal.

While the paper might also be a good fit to a more security-oriented venue, I feel that the demonstration of larger than expected vulnerability of federated learning of multimodal DQA models trained with structured data to gradient inversion will be a useful contribution to the machine learning community and recommend acceptance.